# Steam Explosion-Assisted Extraction of Ergosterol and Polysaccharides from *Flammulina velutipes* (Golden Needle Mushroom) Root Waste

**DOI:** 10.3390/foods13121860

**Published:** 2024-06-13

**Authors:** Wenxin Liu, Jinghua Niu, Fengmei Han, Kai Zhong, Ranran Li, Wenjie Sui, Chao Ma, Maoyu Wu

**Affiliations:** 1State Key Laboratory of Food Nutrition and Safety, College of Food Science and Engineering, Tianjin University of Science & Technology, Tianjin 300457, China; 2Jinan Fruit Research Institute, All-China Federation of Supply & Marketing Co-Operatives, Jinan 250014, China; mch.01@163.com

**Keywords:** steam explosion, *Flammulina velutipes*, extraction, ergosterol, polysaccharides

## Abstract

In this work, steam explosion (SE) was applied to prompt the rapid extraction of ergosterol and polysaccharides from *Flammulina velutipes* root (FVR) waste. Ultrasound-assisted saponification extraction (UASE) followed by water extraction was used to prepare ergosterol and polysaccharides. The results indicated that SE destroyed the complicated structure of FVR and increased its internal porosity and surface roughness. SE caused the thermal degradation of FVR’s structural components and increased the polysaccharide content 0.97-fold. As a result, the extraction yield and efficiency of ergosterol and polysaccharides were improved. The theoretical maximum extraction concentration (*C*_∞_) and diffusion coefficient (*D*) were increased by 34.10% and 78.04% (ergosterol) and 27.69% and 48.67% (polysaccharides), respectively. The extraction yields obtained within 20–30 min of extraction time exceeded those of untreated samples extracted after several hours. For polysaccharides, SE led to a significant reduction in the average molecular weight, increased the percentage of uronic acids and decreased the neutral sugar percentage. The monosaccharide composition was changed by SE, with an increase in the molar ratio of glucose of 64.06% and some reductions in those of other monosaccharides. This work provides an effective method for the processing of fungi waste and adds to its economic value, supporting its high-value utilization in healthcare products.

## 1. Introduction

*Flammulina velutipes*, also known as golden needle mushroom or enokitake, is one of the four most popular edible mushrooms around the world because of its appealing taste, high nutritive value and diverse biological activity [1,2]. It can be regarded as a low-calorie mushroom rich in polyphenols, amino acids (such as ergothioneine), polysaccharides (like *β*-glucans and chitin), terpenoids, vitamins (such as vitamin D_2_) and sterols (like ergosterol) [3,4]. Both the fruiting bodies and the fungal mycelia of *Flammulina velutipes* have been reported to have bioactive polysaccharides, including glucans, which are glucose polymers linked by *α*- and *β*-glycosidic bonds [5]. These polysaccharides exhibit various medical functions, including antioxidant, anti-inflammatory, antitumor, immunomodulatory and hepatoprotective activity [4,6,7]. Moreover, *Flammulina velutipes* can produce ergosterol, a precursor of ergocalciferol, which possesses antioxidant, anti-inflammatory, antitumor and hypocholesterolemic effects, similarly to bioactive phytosterols [8,9,10]. Fungi extracts enriched with ergosterol were reported to not only reduce cholesterol absorption but also restrain its biosynthesis in the human body [5].

Recently, numerous production facilities have been established for the large-scale cultivation of *Flammulina velutipes* in Asian countries, particularly China and Japan, to satisfy the growing demand for processed products like beverages, jellies and noodles [3]. Along with the increasing golden needle mushroom cultivation, this has led to the overproduction of *Flammulina velutipes* root (FVR), which is the main waste after the harvesting of the edible fruiting bodies of fungi. While a small proportion of FVR is utilized as feed, a significant amount is discarded, constituting up to 20% of the total production volume [3,5,11]. Many efforts have been made to raise the golden needle mushroom production yield, while limited research has focused on utilizing FVR waste, which contains valuable nutrients with potential application value in the food and medical industries [3].

To propose an efficient and environmentally friendly process, a novel extraction method involving steam explosion-assisted extraction is investigated for the rapid extraction of ergosterol and polysaccharides from FVR. Steam explosion (SE) is a thermochemical–physical treatment process utilizing water as the reaction and explosion medium to enable the hydrothermal conversion and deconstruction of cell wall components in plant and fungi materials [12]. At the steam cooking stage, the biomass undergoes complex hydrothermal reactions to cut off the long chains of bio-macromolecular components in the cell wall architecture. In the following sudden decompression stage, it undergoes adiabatic flashing to tear the bio-network architecture and release the bioactive products [13,14]. Both of them contribute to the compulsive release and rapid extraction of bioactive products. Compared with conventional procedures, it is more efficient, as it reduces the extraction time and solvent amount and increases the extraction yields. In particular, steam explosion-assisted extraction has been shown to improve the extraction yields of polysaccharides, polyphenols, flavonoids, saponins and alkaloids, etc., from many types of medicinal and edible plants [14,15,16,17,18]. Our previous studies on the SE modification of wheat bran and apple pomace have confirmed that SE can promote the conversion of IDF to SDF and thus increase the extraction yield of SDF [14,15]. Wang et al. studied the disruptive effects of SE on the raspberry leaf structure and a notable 23% increase in the total phenolic content was obtained after SE treatment [16]. Sui et al. reported that the enhanced extraction of saponins from *Radix Astragali* by SE could be attributed to the altered porous plant structure, which improved the solute–solvent accessibility and internal mass transfer during the extraction process [17]. Although the high efficiency of steam explosion-assisted extraction has been confirmed in the extraction of bioactive products from plant materials, its effectiveness when applied to edible mushrooms needs to be further investigated.

In this work, SE was innovatively applied in the value-added utilization of edible mushroom waste, prompting the rapid extraction of ergosterol and polysaccharides from FVR waste. The ultrasound-assisted saponification extraction (UASE) method, followed by the water extraction and alcohol precipitation of steam-exploded FVR, was investigated to prepare ergosterol and polysaccharides, respectively. The microstructural, compositional and thermodynamic properties of FVR before and after SE treatment were systematically compared. The extraction yields and diffusion kinetic rules of ergosterol and polysaccharides from steam-exploded FVR were evaluated, and the structural properties of heterogeneous polysaccharides were characterized, including the main components, monosaccharide composition and molecular weight distribution. This simple procedure, used to valorize FVR waste and manufacture high-value and functional products, is demonstrated to be economical and ecofriendly, providing a new approach for the rapid reduction and efficient utilization of edible mushroom waste.

## 2. Materials and Methods

### 2.1. Materials

*Flammulina velutipes* roots (FVR) were provided by Shandong Youhe Biotechnology Co., Ltd. (Zoucheng, China). All standard chemicals and reagents of analytical grade were purchased from Sinopharm Chemical Reagent Co., Ltd. (Shanghai, China).

### 2.2. Steam Explosion Treatment

The experiment was conducted in an automated laboratory batch reactor (5 L). The FVR sample (300 g) was fed into the reactor and saturated steam was injected as the heating medium. The steaming temperature and explosion pressure were set and maintained at 151.87 °C (0.5 MPa), 184.10 °C (1.0 MPa) and 198.33 °C (1.5 MPa) for 5 min, 7 min and 10 min, with the SE severity (lgR) of 2.23, 3.32 and 3.74, respectively. Subsequently, upon opening the ball valve, the FVR was suddenly exploded into the receiving chamber. The treated FVR samples were then dried in an oven at 60 °C for 8 h and stored at room temperature. Steam explosion (SE) and extraction scheme for ergosterols and polysaccharides from *Flammulina velutipes* root (FVR) waste are demonstrated as Figure 1.

### 2.3. Physicochemical Property Characterization of FVR

The scanning electronic microscopy (SEM) observation of the FVR samples was conducted with a JEOL JSM-6700F system (JEOL Ltd., Tokyo, Japan). Prior to measurement, the sample powders were frozen in liquid nitrogen and dried in a vacuum freeze-dryer. It was then fixed and coated with a thin golden layer using a sputter-coater (Hitachi Science Systems Co., Ltd., Tokyo, Japan). Then, the coated samples were observed and photographed at a 10 kV accelerating voltage, 4.0 spot size, 30 µm objective aperture and 11.9 mm working distance.

The total sugar content was measured using the phenol–sulfuric acid method according to GB/T 15672-2009 [19]; the polysaccharide content was measured using a TU-1810 UV-VIS spectrophotometer (Beijing Puxi General Instrument Co., Ltd., Beijing, China) according to NY/T 1676-2023 [20]; and the protein, lipid and ash content was determined according to the AOAC 920.87, AOAC 983.23 and AOAC 923.03 methods, respectively [21].

The thermogravimetric analysis (TGA) was conducted with a TA instrument (Waters, LLC., New Castle, DE, USA). The FVR sample was located in an aluminum crucible and heated from 30 °C to 600 °C at a rate of 10 °C/min.

### 2.4. Extraction and Determination of Ergosterol from FVR

Ultrasonic extraction: Ultrasound extraction was performed by a KQ-500DE ultrasonic device (Kunshan Ultrasound Instrument Co., Ltd., Kunshan, China). The FVR powder (1 g) was extracted with 30 mL of each selected solvent (n-hexane or 95% ethanol) at 25 °C for 30 min in the ultrasonic device with power of 500 W and a frequency of 40 kHz. The extracts were then filtered and evaporated under reduced pressure to remove the solvent. The residue was dissolved in 10 mL of methanol for ergosterol quantification.

Saponification treatment: The dissolved solution (10 mL) was mixed with 0.1 mol/L vitamin C solution (2 mL) and 2 mol/L potassium hydroxide solution (10 mL). Saponification was conducted by shaking the mixture in a thermostated bath at 120 rpm and 60 °C for 1 h. The cooled mixture was filtered and treated with 5 mL of saturated sodium chloride solution and 10 mL of petroleum ether for 3 min in the vortex mixer 3 times. The organic phase containing ergosterol was collected and evaporated to dryness under reduced pressure. The resulting residue was dissolved in 10 mL of methanol for HPLC analysis.

Ultrasound-assisted saponification extraction (UASE): The FVR powder (1 g) was extracted for 30 min with 30 mL of 95% ethanol solution containing 2 mol/L potassium hydroxide in an ultrasonic device at the same conditions as in the aforementioned ultrasonic extraction. Then, the cooled extracts were treated with 5 mL of saturated sodium chloride solution and 10 mL of petroleum ether for three repetitions. The organic phase was collected, evaporated and dissolved in 10 mL methanol for further analysis.

The extraction runs were analyzed using a mathematical model deduced from Fick’s second law, as described in our previous study [22]. The theoretical maximum extraction content *C*_∞_ and diffusion coefficient *D* for ergosterol were calculated from the model.

Ergosterol quantification: This was conducted using a HPLC apparatus (LC-20A, Shimadzu, Tokyo, Japan) equipped with a diode array detector (DAD). Separation was carried out on a reversed-phase C18 column (250 mm × 4.6 mm) at 35 °C through a dual-wavelength channel of 281 nm. The mobile phase, consisting of 98% (*v*/*v*) methanol, was pumped at a flow rate of 1 mL/min. Standard ergosterol was used to generate the standard calibration curve.

### 2.5. Extraction and Determination of Polysaccharides from FVR

The treated FVR was extracted with distilled water (1:15, *w*/*v*) at 85 °C under gentle stirring for 2 h and the extraction liquid was concentrated using a RE-52AA vacuum rotary evaporator (Shanghai Yarong Biochemical Instrument Co., Ltd., Shanghai, China). Subsequently, four volumes of anhydrous ethanol were added and left to stand overnight at 4 °C. The mixture was centrifuged at a speed of 3500 r/min for 15 min. The obtained precipitate after centrifugation was deproteinized three times using Sevag reagent and re-dissolved in hot distilled water. The FVR solution was then dialyzed using a 1000 Da dialysis membrane against distilled water and freeze-dried to obtain the crude polysaccharide powder. The extraction yield of polysaccharides in FVR was determined using the gravimetric method. 

The extraction runs were analyzed using a mathematical model deduced from Fick’s second law, as described in our previous study [22]. The theoretical maximum extraction content *C*_∞_ and diffusion coefficient *D* of polysaccharides were calculated from the model.

### 2.6. Structural Characterization of Polysaccharides from FVR

The total sugar content was determined using the phenol–sulfuric acid method, and the uronic acid content was determined by the carbazole and sulfuric acid spectrophotometric method [15]. The protein and ash content was measured according to the methods in Section 2.3.

The monosaccharide composition was determined according to our previous work, with slight modifications [15]. A 10 mg sample was hydrolyzed in 2 mol/L TFA at 110 °C for 3 h. The mixture was treated with repeated methanol to remove the excess TFA. Then, the sample was reduced by NaBH_4_ and acidified with acetic acid. The alditol acetate was acetylated with pyridine–acetic anhydride (1:1) at 105 °C for 1 h. Afterwards, it was determined using an Agilent 7890 A gas chromatography (GC) (Agilent Technologies, Santa Clara, CA, USA) system equipped with an OV-1701 (0.32 mm × 0.5 μm × 30 m) capillary column (Lanzhou Zhongke Kaidi Chemical New Technology Co., Ltd., Lanzhou, China). The column temperature was maintained at 150 °C for 1 min, raised to 200 °C for 10 min at a heating rate of 10 °C/min, raised to 220 °C for 5 min at a heating rate of 5 °C/min and eventually increased to 240 °C for 20 min at a heating rate of 1.5 °C/min. The injector temperature and FID detector temperature were set to 240 °C and 280 °C, respectively. 

The molecular weight distribution of the polysaccharides was determined by a Waters HPLC instrument (RID-10A detector) (Waters Corporation, Milford, MA, USA) equipped with a Shodex SB-805 HQ (8.0 mm × 300 mm) chromatography column (Showa Denko, Tokyo, Japan). The sample was eluted with RO water at a flow rate of 0.8 mL/min and the injection volume was 10 µL. Dextrans of various molecular weights were taken as standards.

### 2.7. Statistical Analysis

Unless otherwise specified, 3 independent tests were performed, and the values are presented as the arithmetic mean ± standard deviation (SD). Statistical differences were calculated by ANOVA and Duncan’s multiple range test. Values were considered significantly different at *p* < 0.05 (SPSS for Window 25.0).

## 3. Results and Discussion

### 3.1. Effect of Steam Explosion on Microstructure, Chemical Composition and Thermodynamic Properties of FVR

The microscopic surface morphology of FVR before and after SE treatment is shown in Figure 2. As depicted, the untreated FVR samples present a complete and orderly surface structure with the dense and tight arrangement of the mycelia. After SE processing, the surface roughness of FVR is increased and the lamellar structure of the cross-section becomes separated. With the increment in the SE severity, the cracks among the mycelial filaments are widened and the physical tearing effect becomes more apparent. At the steam cooking stage, the pressurized and saturated steam triggers a series of thermal–chemical reactions, resulting in the softening of the fungal structure and the degradation of some biomacromolecules that constituted the fungal cell walls [23]. At the instantaneous explosion stage, the steam that has penetrated into the mycelium tissue expands adiabatically, and some condensed water undergoes flash evaporation, significantly destroying the three-dimensional structure of the FVR. Both of these cause the disintegration of the lamellar structure and increase the porosity of the internal structure and the roughness of the surface structure of FVR. The loose mycelial bundles formed after SE facilitate the dissolution and release of the functional components contained within the FVR.

The chemical composition of FVR varies with the SE severity, as listed in Table 1. In this work, the FVR is primarily composed of carbohydrate substances, with total sugar content accounting for 60.22 ± 0.47%, of which polysaccharides account for 3.92 ± 0.04%, followed by protein, ash and fat, accounting for 20.98 ± 0.21%, 8.24 ± 0.07% and 4.79 ± 0.52%, respectively. Compared to the untreated samples, the total sugar content in the steam-exploded FVR is slightly decreased, while the polysaccharide content is significantly increased. As the SE severity increases, the total sugar content decreases to 56.52 ± 0.12%. This may be due to SE hydrolyzing some polysaccharides, such as chitin and glucans, into monosaccharides and oligosaccharides, leading to an increment in water-soluble sugars, which can be removed by the water-washing step after SE treatment. Moreover, some saccharides may undergo Maillard reactions with protein and amino compounds to form biopolymer aggregates, also leading to a reduction in the total sugar content. After SE treatment, the polysaccharide content is firstly increased and then decreased, reaching a maximum of 7.71 ± 0.15% at the SE severity of lgR = 3.32, which is increased by 96.68% compared to the untreated FVR. On one hand, SE caused the breakage of some glycosidic bonds in the cell wall polysaccharides such as chitin and glucans, resulting in the transformation of structural polysaccharides into amorphous and soluble polysaccharides. On the other hand, SE improved the FVR’s porous structure, which was beneficial to the dissolution and release of polysaccharides [13,24]. The reduction in polysaccharide content at the higher SE severity may be due to the excess degradation of polysaccharides into oligosaccharides or monosaccharides, which cannot be precipitated with an ethanol solution during the extraction process [16]. It could also be caused by the high-temperature carbonization of the polysaccharides. The fat and protein content of FVR is lowered after SE treatment, indicating that the high-pressure and -temperature reaction conditions can induce the thermal denaturation and degradation of the proteins and the decomposition of the lipids into free fatty acids.

The thermogravimetric and differential thermogravimetric curves of FVR under different SE conditions are depicted in Figure 3, and the thermal decomposition characteristics are presented in Table 2. The thermal decomposition process of FVR can be divided into three main stages according to the horizontal tangent preceding each peak in the DTG curve: the first stage of dehydration occurred in the range of 30–150 °C, with a weight loss rate of 8.46% for the raw samples and 8.40–10.70% for the steam-exploded FVR; the second stage of carbohydrate polymer and protein decomposition was located in the range of 150–400 °C, with a weight loss rate of 70.72% for the raw samples and 42.28–56.16% for the steam-exploded FVR; and the third stage of char formation occurred in the range of 400–600 °C, with a weight loss rate of 10.81% for the raw samples and 11.19–13.80% for the steam-exploded FVR. All samples experienced a small amount of mass loss in the first stage, owing to the water evaporation. There was no significant difference in the weight loss rates during the first stage, indicating that the amounts of free water and bound water in FVR were similar before and after SE treatment. The sharp decline in the TG curve during the second stage is mainly caused by the thermal decomposition of the major organic matter in FVR, with two distinct peaks (255 °C and 296 °C) appearing in this decomposition process. The DTG peak located at 255 °C can be attributed to the thermal degradation of carbohydrates, and the peak at 296 °C corresponds to the degradation of nitrogen-containing compounds [25,26]. As the SE severity increased, the peak height of the above two peaks significantly decreased, leading to a large reduction in weight loss. This is because SE can enhance the degradation of the characteristic functional groups of saccharides, proteins and lipids, such as carbonyl, hydroxyl and carboxyl groups, which means that FVR after SE has higher thermal stability. The third stage is the slow decomposition process of the charred residues. The char, ash and fixed carbon remaining from the thermal decomposition of organic components undergo carbonization reactions under high temperatures [27]. The total weight loss of the steam-exploded FVR is much higher than that of the untreated FVR samples, indicating that SE treatment improves the thermal stability of FVR. However, there is no significant difference among the steam-exploded FVR samples under different severities.

### 3.2. Effect of Steam Explosion on Extraction of Ergosterol from FVR 

Ergosterol has a dual-molecule structure and is soluble in solvents with some polarity. The effects of two commonly used extraction solvents (ethanol and n-hexane, with ethanol being more polar than n-hexane) on the extraction yields of ergosterol were compared [5,28]. As demonstrated in Figure 4, the yields of ergosterol extracted using ethanol and n-hexane as solvents were determined to be 1.35 ± 0.07 mg/g and 1.20 ± 0.05 mg/g, respectively. The yield of ergosterol using 95% (*v*/*v*) ethanol as the extraction solvent was significantly higher than that using n-hexane as an extraction solvent (*p* < 0.01), and ethanol was easy to recover with less pollution. Meanwhile, n-hexane, as an extraction solvent, had a lower extraction yield for ergosterol but presented relatively fewer impurities in the extracts, owing to its lower polarity. Therefore, 95% (*v*/*v*) ethanol was selected as the extraction solvent for ergosterol. Further comparison was performed on the influence of different extraction methods on the extraction yields of ergosterol. The order of the ergosterol yields obtained by different extraction methods was as follows: UASE (1.69 ± 0.07 mg/g) > ultrasound extraction followed by saponification treatment (1.53 ± 0.05 mg/g) > ultrasound extraction (1.37 ± 0.06 mg/g). Clearly, the yield of ergosterol extracted by the UASE method was significantly higher than that for the other two methods (*p* < 0.01). Saponification is a hydrolysis reaction whereby the hydroxide can break the ester bonds between fatty acids and glycerol in triglycerides, converting conjugated ergosterol into free ergosterol. This promoted the rapid dissolution and dispersion of ergosterol from the mycelium cells into the solvent, thereby increasing the extraction yield of ergosterol. The high yield of ergosterol obtained by the UASE method may have been due to the cavitation effect generated by the ultrasound waves, which accelerated the breakdown of the fungal cell wall and promoted the saponification reaction [5]. Therefore, through process optimization, the 95% ethanol solution was chosen as the extraction solvent, and the UASE method was used to prepare ergosterol.

The extraction kinetics of ergosterol under the optimized extraction process was further investigated. As demonstrated in Figure 5, the extraction yield of ergosterol gradually increased with the prolongation of the extraction time. The extraction yield of ergosterol prepared from untreated FVR reached its maximum value at about 80 min of extraction time, while the extraction yield of steam-exploded FVR reached its highest value at around 20 min of extraction time. The theoretical maximum extraction concentration (*C*_∞_) and diffusion coefficient (*D*) of ergosterol were deduced from the Fick model [19]. Compared with the untreated samples, the *D* value and *C*_∞_ value for steam-exploded FVR were increased by 78.04% and 34.10%, respectively. This indicates that SE can not only increase the extraction yield of ergosterol but also shorten the extraction time, allowing the ergosterol in FVR to be fully extracted in a short period of time. SE significantly disrupted the multi-scale structure of the mycorrhizae, resulting in an increase in the internal pore size and number, thereby enhancing the connectivity of the porous network. This action should be beneficial to improve the solvent accessibility and mass transfer efficiency, making it easier for the extraction solvent to reach the extraction sites inside the porous structure [19]. The standard calibration curve of ergosterol and the HPLC chromatography results for the standard ergosterol and ergosterol prepared from steam-exploded FVR are demonstrated in Appendix A.

### 3.3. Effect of Steam Explosion on Exaction of Polysaccharides from FVR

Figure 6 presents the extraction kinetic curves and parameters of the polysaccharides from FVR before and after SE treatment. As depicted, the extraction yield of polysaccharides from untreated FVR gradually increased with the prolonged extraction time, eventually reaching a maximum value of 3.90% after 4 h of extraction. The extraction yield of polysaccharides from the steam-exploded FVR exceeded the final polysaccharide yield of the untreated FVR samples after only 0.5 h of extraction. Under the SE conditions of an explosion pressure of 1.1 MPa and retention time of 7 min, the *C*_∞_ of polysaccharides was calculated to be 78.40 mg/g, which was an increase of 27.69% compared to that of the raw materials; the mass transfer coefficient (*D*) was 30.150·10^−11^ m^2^/s, which was 0.49 times higher than that of the raw materials. These findings indicate that SE can significantly promote the dissolution and release of polysaccharides from FVR. One reason for this is that the physical explosion effect produced during the instantaneous decompression stage can destroy the dense lamellar structure inside the mycelia, thus exposing the internal structure and increasing the porosity and specific surface area [29]. The destruction effects of SE on the mycelia’s multi-scale structure may enhance the accessible surface area for the solvent to establish contact with the solute and open up the channels for the release of the solute, leading to a reduction in the mass transfer resistance and an increment in the mass transfer coefficient, finally promoting the dissolution of the polysaccharides. The other reason is that the hydrothermal reactions that occur during the SE process may induce the thermal degradation of the structural macromolecules within FVR. Certain components, such as chitin and glucan, that constitute the cell walls are prone to melting and hydrolyzing under thermal and acidic conditions. The hydrogen bonds and glycosidic bonds within these molecules are easily broken at high temperatures, thereby transforming structural polymers into water-soluble substances [13]. This may cause a some insoluble dietary fiber to be converted into soluble dietary fiber, which significantly increased the theoretical maximum extraction concentration (*C*_∞_) of the polysaccharides in the steam-exploded FVR [17]. Therefore, the increment in the polysaccharide content after SE treatment can be attributed to the improved dissolution of the polysaccharides, induced by structural changes and the partial transformation of insoluble dietary fiber.

### 3.4. Effect of Steam Explosion on Structural Properties of Polysaccharides from FVR

The chemical composition of the polysaccharides prepared from FVR before and after SE treatment is presented in Table 3. The polysaccharides in all samples were mainly composed of neutral sugars, ranging from 76.32 ± 0.01% to 86.26 ± 0.05%, and also contained 4.22 ± 0.06%~5.20 ± 0.19% protein, 1.61 ± 0.10%~2.90 ± 0.16% uronic acids and 1.60 ± 0.23%~2.01 ± 0.12% ash, on a dry weight basis. With the increment in the SE severity, the content of neutral sugars gradually decreased. The polysaccharides from untreated FVR had many side chains with neutral sugars, and SE led to the partial breakage of the side chains, transforming them into dietary fiber with a low polymerization degree, oligosaccharides and small-molecule substances, which contributed to the reduction in the neutral sugar content [15]. Under the high-pressure and thermal–acidic environment during SE processing, the glycosidic bonds of uronic acid were less prone to acid hydrolysis compared to the neutral sugars, resulting in a relative increase in the uronic acid content in the polysaccharides. Additionally, the increase in uronic acid content might have been due to the auto-hydrolysis of polysaccharides during the SE process, generating some monosaccharides containing uronic acid structures, which could react with carbazole to form purple compounds, leading to a relatively high measured value. These results indicate that SE processing can alter the major chemical composition of the polysaccharides in FVR, as presented by the reduced content of neutral sugars and the raised content of uronic acids.

The gas chromatography profiles and monosaccharide compositions of the polysaccharides prepared from FVR before and after SE are shown in Figure 7 and Appendix A and Table 4. The polysaccharides of all FVR samples mainly consisted of five monosaccharides, glucose, galactose, mannose, rhamnose and arabinose, which was in accordance with other work [4]. The monosaccharide composition of the polysaccharides changed significantly after SE treatment. The highest molar ratio of glucose in the polysaccharides from steam-exploded FVR reached 43.85 mol%, with an increase of 64.06% compared to the polysaccharides from untreated FVR. This is primarily because the SE process can cause the thermal degradation of the cell wall components, such as chitin and glucan, resulting in more soluble components that can be easily extracted into polysaccharides, thereby increasing the polysaccharide content and the proportion of these saccharides within the polysaccharides. Compared to the polysaccharides from untreated samples, the molar ratios of rhamnose, arabinose, mannose and galactose in the polysaccharides from the steam-exploded FVR were decreased by a maximum of 55.05%, 40.17%, 53.05% and 71.33%, respectively. This could have been due to the fact that the higher proportion of glucose in the polysaccharides led to relatively lower molar ratios of other saccharides. Moreover, SE may cause the decomposition of polysaccharides containing these monosaccharides, which makes them difficult to precipitate with ethanol, also contributing to the lowered molar ratios of galactose, mannose, rhamnose and arabinose.

The molecular weight distribution of the polysaccharides from FVR before and after SE is depicted in Figure 8 and Appendix A and Table 5. The polysaccharides mainly contain four different molecular weight fractions. This indicates that the prepared polysaccharides should be composed of highly dispersed, heterogeneous polysaccharide fractions. For the polysaccharides from untreated FVR, peak 1, with a number-average molecular weight (*M*n) of 2.12 × 10^6^ Da, accounts for the largest peak area percentage (66.40%). Under the severest SE conditions (explosion pressure 1.5 MPa and retention time 7 min), the *M*n corresponding to peak 1 in polysaccharides is decreased to 1.54 × 10^6^ Da, where the weight-average molecular weight (*M*w) reduced to 2.04 × 10^6^ Da and the relative peak area percentage decreased to 52.15%. The peak area percentages of other lower-molecular-weight fractions (peaks 2, 3, and 4) increased, especially peak 4, which rose from 17.87% to 34.04%. These results suggest that SE can cause the transformation of high-molecular-weight polysaccharides to low-molecular-weight polysaccharides, resulting in a reduction in the overall molecular weight distribution [15]. This may be due to two reasons: firstly, the thermal degradation or acid-like decomposition of polysaccharides in the intrinsic polysaccharides, such as gums, occurs during the SE process of FVR roots, resulting in a reduction in their molecular weights; secondly, SE could destroy the structural polysaccharides in the cell walls of FVR by weakening the glycosidic bonds and hydrogen bonds within and between the polysaccharide chains, causing the destruction of their spatial structure and the cleavage of the molecular chains [17]. Both of these contributed to the reduced degree of polymerization and the lowered molecular weight of the polysaccharides. The changes in the molecular weight distribution of polysaccharides would help to improve their extraction performance and physiological function.

## 4. Conclusions

In this study, an exploratory application of SE for the minimization and valorization of FVR waste was established to prepare ergosterol and polysaccharides with high extraction yields and efficiency. The instantaneous explosion effect of the SE process destroyed the dense structure of FVR, increasing its internal porosity and surface roughness, as presented by the SEM images. The hydrothermal–chemical effect of the SE process caused the thermal degradation of some structural components of FVR, reducing the content of total sugars, lipids and proteins, while increasing the polysaccharide content 0.97-fold in comparison with the untreated samples and also enhancing the thermal stability of the FVR. The above compositional and structural changes induced by SE treatment facilitated the dissolution and release of ergosterol and polysaccharides from FVR, improving the extraction yield and efficiency of ergosterol and polysaccharides. After SE, the theoretical maximum extraction concentration (*C*_∞_) and diffusion coefficient (*D*) of ergosterol and polysaccharides were increased by 34.10% and 78.04% (for ergosterol) and 27.69% and 48.67% (for polysaccharides). The extraction yields obtained within 20–30 min of extraction time exceeded those of untreated samples extracted after several hours. For FVR processed by SE treatment, there was a significant reduction in the *M*w and *M*n of heterogeneous polysaccharides. The percentage of uronic acids increased, while the percentage of neutral sugars decreased in the FVR polysaccharides. The monosaccharide composition was also changed after SE treatment, with an increase in the molar ratio of glucose of 64.06% and a decrease in the molar ratios of galactose, mannose, rhamnose and arabinose. This work not only provides an effective method for the processing of fungi waste but also adds to its economic value; the results can be applied in the research and development of healthcare products. Our next work will focus on the separation and purification of polysaccharides from SE-treated FVR waste and their application to the advanced manufacturing of flexible composite functional films with superior electrical conductivity.

## Figures and Tables

**Figure 1 foods-13-01860-f001:**
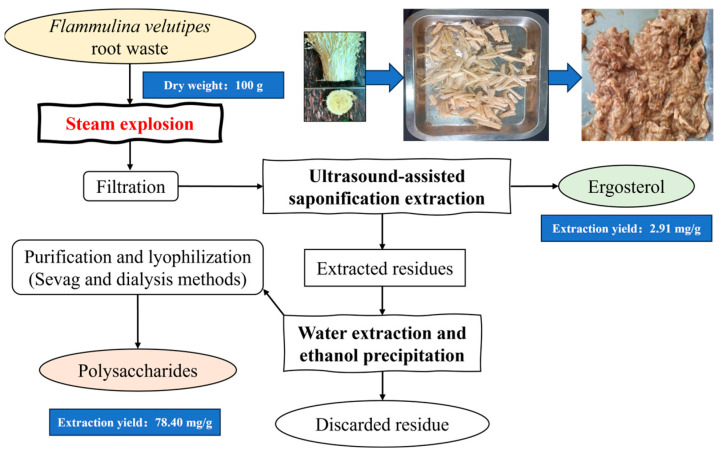
Steam explosion (SE) and extraction scheme for ergosterols and polysaccharides from *Flammulina velutipes* root (FVR) waste.

**Figure 2 foods-13-01860-f002:**
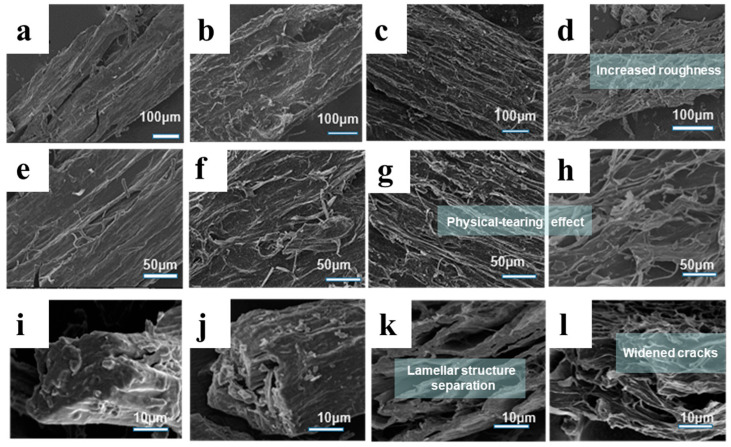
The scanning electronic microscopy (SEM) images of *Flammulina velutipes* roots (FVR) before and after steam explosion (SE) ((**a**,**e**,**i**): untreated FVR ×200, ×500, ×1000; (**b**,**f**,**j**): steam-exploded FVR at lgR = 2.23 ×200, ×500, ×1000; (**c**,**g**,**k**): steam-exploded FVR at lgR = 3.32 ×200, ×500, ×1000; (**d**,**h**,**l**): steam-exploded FVR at lgR = 3.74 ×200, ×500, ×1000).

**Figure 3 foods-13-01860-f003:**
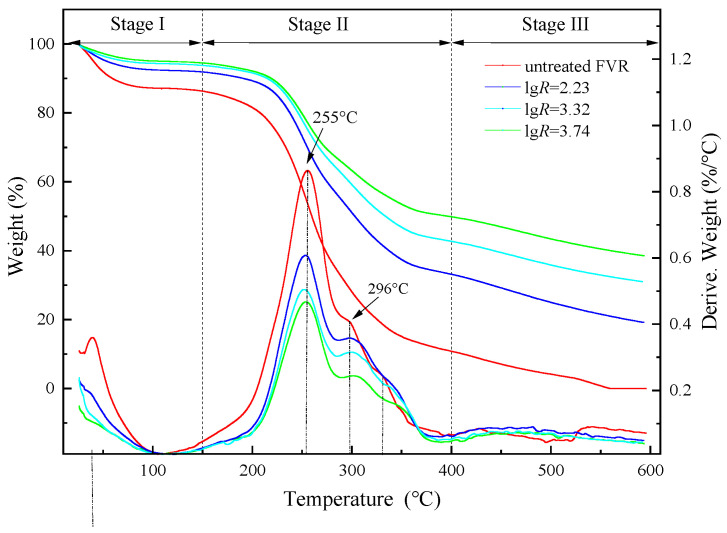
Thermal gravimetry (TG) and differential thermal gravimetry (DTG) curves of *Flammulina velutipes* roots (FVR) before and after steam explosion (SE).

**Figure 4 foods-13-01860-f004:**
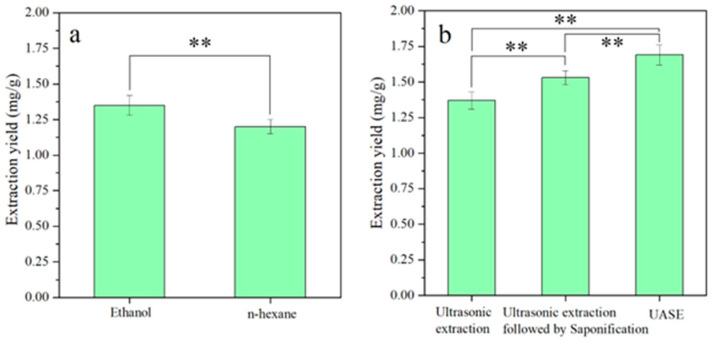
Extraction yields of ergosterol extracted by different solvents (**a**) and different methods (**b**). **: significant difference, *p* < 0.01.

**Figure 5 foods-13-01860-f005:**
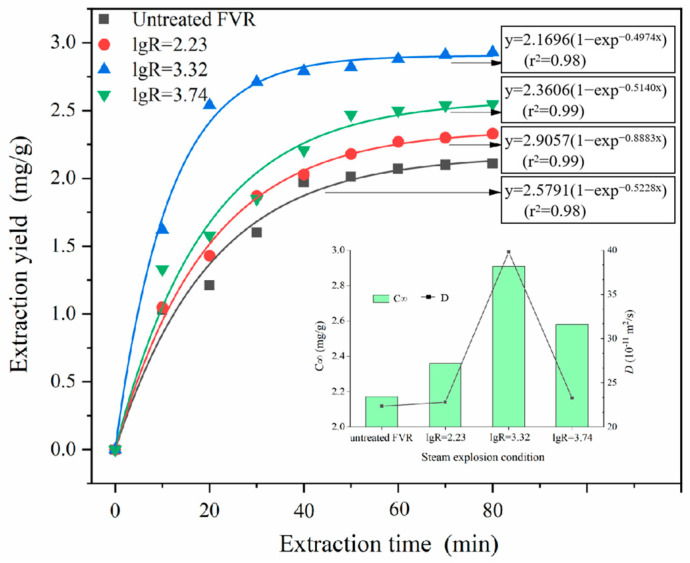
Extraction kinetic curves and kinetic parameters (*C*_∞_ and *D*) of ergosterol from *Flammulina velutipes* roots (FVR).

**Figure 6 foods-13-01860-f006:**
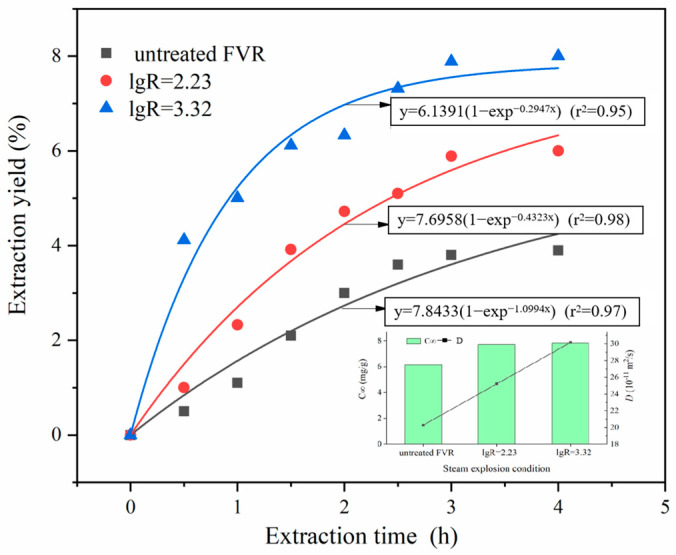
Extraction kinetic curves and kinetic parameters (*C*_∞_ and *D*) of polysaccharides from *Flammulina velutipes* roots (FVR).

**Figure 7 foods-13-01860-f007:**
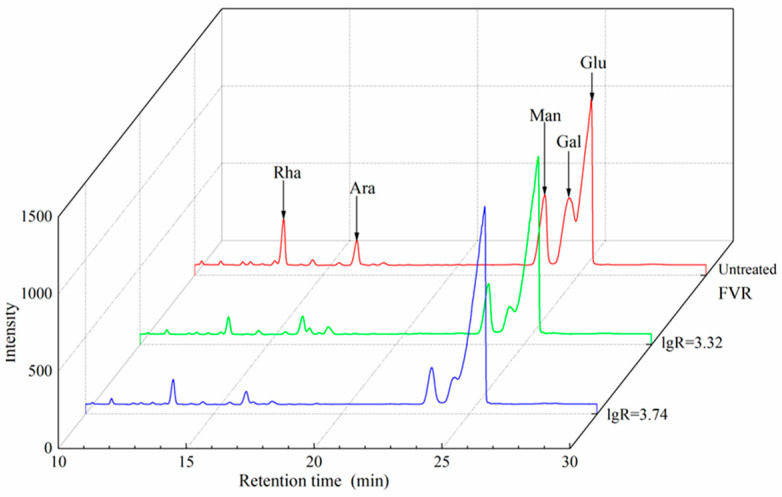
GC chromatography of polysaccharides from *Flammulina velutipes* roots (FVR).

**Figure 8 foods-13-01860-f008:**
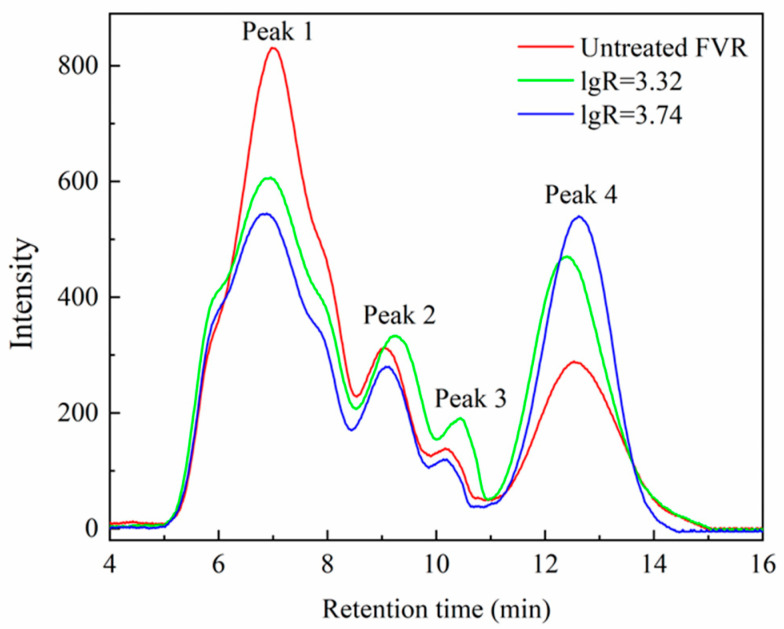
Molecular weight distribution of polysaccharides from *Flammulina velutipes* roots (FVR).

**Table 1 foods-13-01860-t001:** Chemical composition of *c* before and after steam explosion (SE).

SE Severity	Total Sugar Content (%)	Polysaccharide Content (%)	Lipid Content (%)	Protein Content (%)	Ash Content (%)
Untreated FVR	60.22 ± 0.47 ^a^	3.92 ± 0.04 ^a^	4.79 ± 0.52 ^a^	20.98 ± 0.21 ^a^	8.24 ± 0.07 ^bc^
lgR = 2.23	59.93 ± 0.43 ^a^	5.92 ± 0.14 ^a^	4.44 ± 0.45 ^ab^	20.48 ± 0.19 ^ab^	8.57 ± 0.14 ^b^
lgR = 3.32	57.13 ± 0.33 ^ab^	7.71 ± 0.15 ^b^	4.28 ± 0.29 ^ab^	19.34 ± 0.14 ^bc^	8.85 ± 0.32 ^b^
lgR = 3.74	56.52 ± 0.12 ^b^	7.65 ± 0.16 ^c^	4.26 ± 0.02 ^c^	18.19 ± 0.48 ^c^	9.06 ± 0.06 ^a^

Results are given as mean ± standard deviation. Different letters in the same column indicate significant differences (*p* < 0.05).

**Table 2 foods-13-01860-t002:** Weight loss and DTGmax of *Flammulina velutipes* roots (FVR) before and after steam explosion (SE).

SE Severity	Stage I		Stage II		Stage III	
	Weight Loss%	DTGmax %/°C	Weight Loss %	DTGmax %/°C	Weight Loss %	DTGmax %/°C
Untreated FVR	8.46	0.36	70.72	0.86	10.81	0.42
lgR = 2.23	7.85	0.24	56.16	0.61	11.19	0.36
lgR = 3.32	10.70	0.25	48.86	0.50	11.63	0.31
lgR = 3.74	8.40	0.15	42.28	0.47	13.80	0.24

**Table 3 foods-13-01860-t003:** Chemical composition of polysaccharides from *Flammulina velutipes* roots (FVR).

Component	Untreated FVR	lgR = 3.32	lgR = 3.74
Neutral sugar%	86.26 ± 0.05 ^ab^	85.00 ± 0.01 ^a^	76.32 ± 0.01 ^b^
Uronic acid%	1.61 ± 0.10 ^b^	2.72 ± 0.41 ^a^	2.90 ± 0.16 ^a^
Protein%	4.65 ± 0.42 ^a^	4.22 ± 0.06 ^b^	5.20 ± 0.19 ^a^
Ash%	1.83 ± 0.32 ^b^	1.60 ± 0.23 ^b^	2.01 ± 0.12 ^a^

Results are given as mean ± standard deviation. Different letters in the same column indicate significant differences (*p* < 0.05).

**Table 4 foods-13-01860-t004:** Monosaccharide composition (molar ratio) of polysaccharides from *Flammulina velutipes* roots (FVR).

Monosaccharide	Retention Time (min)	Molar Ratio (mol%)		
		Untreated FVR	lgR = 3.32	lgR = 3.74
Rhamnose	13.57	3.27	1.33	1.47
Arabinose	15.32	2.34	1.90	1.40
Mannose	24.14	8.84	6.23	4.15
Galactose	25.12	13.22	4.52	3.79
Glucose	25.74	26.74	39.52	43.85

**Table 5 foods-13-01860-t005:** Number-average molecular weight (*M*n) and weight-average molecular weight (*M*w) of polysaccharides from *Flammulina velutipes* roots (FVR).

Steam Explosion Condition	Peak	Time/min	*M*n/Da	*M*w/Da	Polydispersity Index	Relative Peak Area/%
Untreated FVR	1	6.97	2.12 × 10^6^	2.73 × 10^6^	1.29	66.40
	2	9.04	7.86 × 10^4^	1.62 × 10^5^	2.05	12.50
	3	10.161	4.89 × 10^3^	6.89 × 10^4^	1.40	3.23
	4	12.52	1.01 × 10^3^	1.72 × 10^4^	1.70	17.87
lgR = 3.32	1	6.954	1.60 × 10^6^	2.07 × 10^6^	1.25	54.12
	2	9.26	5.23 × 10^4^	1.27 × 10^5^	2.43	14.39
	3	10.44	3.45 × 10^3^	5.84 × 10^3^	1.69	4.64
	4	12.40	2.32 × 10^3^	2.54 × 10^4^	1.09	26.86
lgR = 3.74	1	6.89	1.54 × 10^6^	2.04 × 10^6^	1.32	52.15
	2	9.12	7.83 × 10^4^	1.67 × 10^5^	2.13	11.84
	3	10.19	6.05 × 10^3^	7.35 × 10^3^	1.21	1.97
	4	12.62	1.89 × 10^3^	2.02 × 10^4^	1.07	34.04

## Data Availability

The original contributions presented in the study are included in the article/Appendix A, further inquiries can be directed to the corresponding authors.

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
