# Peer review of "Steam Explosion-Assisted Extraction of Ergosterol and Polysaccharides from Flammulina velutipes (Golden Needle Mushroom) Root Waste"

_foods, 2024, doi:10.3390/foods13121860_

Round 1
Reviewer 1 Report
Comments and Suggestions for Authors
Minor remarks
Avoid using abbreviations in the abstract, but, if necessary, define them properly.
Use the abbreviated units. For instance, instead of minutes and hours, use „min“ and „h“.
The abbreviation “UASE” can be defined for the following term: “Ultrasound-Assisted Saponification Extraction”. Define the abbreviations and use them in the manuscript. Also, give the explanation of used abbreviations in the presented tables.
The resolution of Figures 4 and 5 should be improved.
Major remarks
I noticed a wide range of cited references in the Introduction section. Avoid just piling on references without deep discussion in the manuscript. It is advisable to discuss each reference separately.
The novelty and main contribution of this study is not well presented so it should be better highlighted.
I recommend giving the schematic representation of this research for easier understanding the whole concept.

The English language is acceptable and understandable.
Reviewer 2 Report
Comments and Suggestions for Authors
my comments are attached to the file.

Minor editing of English language required.
Round 2
Reviewer 1 Report
Comments and Suggestions for Authors
The manuscript can be accepted for publication in the present form.
Comments on the Quality of English LanguageThe English language is acceptable.
Author Response
Dear editor and reviewers,
We would like to thank you for giving us a chance to re-revise the paper (“Steam explosion-assisted extraction of ergosterol and polysaccharides from Flammulina velutipes (golden needle mushroom) root waste”, foods-3022891), and also thank editor and reviewers very much for giving us constructive suggestions which would help us to improve the quality of the paper. We have checked the manuscript and modified it according to the editor’s comments. Revised portion are marked in Red in the paper. The main corrections in the paper (in Red) and the response to the editor’s comments (in Blue) are as following:
The editor’s Comments: The present research article showed a high percentage of self-citation (27%). In my opinion the total amount of self-citation should not exceed 15%. My suggestion is to ask the authors to reduce the amount of self-citations.
The authors’ Answer: Thanks for your kind suggestion. According to editor’s suggestion, we have changed some references to reduce the self-citation to 15%. The added references are listed as below in the re-revised manuscript.
12. Zhang, Y.; Feng, Y.; Shi, H.; Ding, K.; Zhou, X.; Zhao, G.; Hadiatullah, H. Impact of steam explosion pretreatment of defatted soybean meal on the flavor of soy sauce. LWT.2022, 156, 113034.
13. Wang, L.; Pang, T.; Kong, F.; Chen, H. Steam explosion pretreatment for improving wheat bran extrusion capacity. Foods. 2022, 11, 2850.
18. Liang, Y.; Yang, Y.; Zheng, L.; Zheng, X.; Xiao, D.; Wang, S.; Ai, B.; Sheng, Z. Extraction of pectin from passion fruit peel: composition, structural characterization and emulsion stability. Foods. 2022, 11, 3995.
The Reviewers’ Comments: Minor editing of English language required.
The authors’ Answer: We apologize for any linguistic inaccuracies in our manuscript. We have engaged a native English speaker to assist in refining the article. We trust that these efforts have significantly improved the flow and language quality of the revised manuscript. We extend our gratitude for your dedicated efforts on this paper and appreciate the valuable feedback provided by the reviewers.
Once again, we appreciate for your and all the reviewers’ valuable comments and suggestions earnestly, and hope that the corrections will meet with approval.
Your sincerely,
Wenxin Liu, Jinghua Niu, Fengmei Han, Kai Zhong, Ranran Li, Wenjie Sui, Chao Ma, Maoyu Wu
Reviewer 2 Report
Comments and Suggestions for Authors
accept
Comments on the Quality of English Languageminor
Author Response

(The authors gave the same response as above.)
